# N6-Methyladenosine-Related lncRNA Signature Predicts the Overall Survival of Colorectal Cancer Patients

**DOI:** 10.3390/genes12091375

**Published:** 2021-08-31

**Authors:** Wei Song, Jun Ren, Wenzheng Yuan, Rensheng Xiang, Yuhang Ge, Tao Fu

**Affiliations:** Department of Gastrointestinal Surgery II, Renmin Hospital of Wuhan University, Wuhan 430060, China; whdxsw@whu.edu.cn (W.S.); renjun0414@163.com (J.R.); yuanwz2013@163.com (W.Y.); 2014302180192@whu.edu.cn (R.X.); 2013302180234@whu.edu.cn (Y.G.)

**Keywords:** colorectal cancer, N6-methyladenosine, long non-coding RNA, prognostic model

## Abstract

Background: The N6-methyladenosine (m6A) RNA modification can modify long non-coding RNAs (lncRNAs), thereby affecting the tumorigenesis and progression of tumors. However, the underlying role of m6A-modified lncRNAs in colorectal cancer (CRC) remains largely unknown. Therefore, our aim was to assess the prognostic value of m6A-modified lncRNAs in CRC patients. Methods: The gene expression and clinicopathological data of CRC were extracted from The Cancer Genome Atlas (TCGA) database. Pearson correlation analysis was used to investigate the m6A-modified lncRNAs. Consensus clustering was conducted to identify molecular subtypes of CRC, and the clinical significance of molecular subtypes was identified. The least absolute shrinkage and selection operator analysis (LASSO) was applied to establish a risk signature. Finally, a prognostic nomogram with risk score and clinicopathological variables was established. Results: In total, 29 m6A-modified lncRNAs were identified as prognostic lncRNAs. Two molecular clusters were identified and significant differences were found with respect to clinicopathological features and prognosis. Cluster1 is associated with poor overall survival (OS), down-regulation of Programmed cell death ligand-1 (PD-L1) expression, lower immune score, and less immune cell infiltration. Then, an m6A-modified lncRNA signature for predicting OS was constructed in the TCGA training cohort. The signature demonstrated favorable prediction performance in both training and validation sets. Compared with low-risk patients, patients with high risk showed worse clinical outcomes, lower immune scores, and downregulated PD-L1 expression. Further analysis indicated that the signature was an independent prognostic indicator, and then a prognostic nomogram based on risk score, tumor location, and tumor stage was established. Conclusions: Our study identified a seven m6A-modified lncRNA signature and established a prognostic nomogram that reliably predicts OS in CRC. These findings may improve the understanding of m6A modifications in CRC and provide insights into the prognosis and treatment strategy of CRC.

## 1. Introduction

Colorectal cancer (CRC) is the third most common cancer in the world and the second leading cause of mortality worldwide. Approximately 1.8 million new cases occur worldwide each year, and approximately 900,000 people die of CRC [1,2]. In the last decade, progress has been made in the diagnosis and treatment of CRC, the mortality of CRC remains high due to the lack of biomarkers for early screening and prognosis prediction, meaning that many cases are not diagnosed until advanced clinical stages [2]. Therefore, efficient prognostic biomarkers and functional signatures may be beneficial to realize individualized survival predictions and provide patients with an optimal therapeutic approach.

N6-methylandenosine (m6A) modification, the most common posttranscriptional modification of mRNAs and non-coding RNAs, plays a vital role in RNA maturation, export, stability, translation, export, and decay [3,4]. To date, m6A modifications were identified in more than 7600 genes and 300 non-coding RNAs in mammals [5]. As an invertible and dynamic RNA epigenetic process, molecular components of m6A RNA methylation involves intracellular methyltransferases (“writers”), demethylases (“erasers”), and signal transducers (“readers”), which regulate gene expression and are associated with various biological functions, such as RNA splicing, export, stability, translation, and the biogenesis of ncRNAs [6,7]. Accumulating evidence shows that changes in m6A modification patterns are related to tumorigenesis and the progression of various types of cancer [8,9,10]. Abnormal m6A methylation levels can affect the regulation of self-renewal of cancer stem cells and tumor immune response, and microRNA (miRNA) editing, promotion of cancer cell proliferation, and resistance to radiotherapy or chemotherapy [11,12,13,14]. For example, the m6A modification mediated CBX8 interacts with KMT2b and Pol II to promote LGR5 expression, which contributes to increase the stemness of cancer and reduce the chemosensitivity of colon cancer [13]. Han et al. [14] revealed that METTL3-mediated m6A modification can promote the proliferation of bladder cancer by promoting pri-miR221 and pri-miR222 maturation.

Long noncoding RNAs (lncRNAs) are defined as non-protein-coding RNA transcripts of over 200 nucleotides and lacking a canonical open reading frame (ORF), and they have been reported to be abnormally expressed in various types of cancer cells and play critical regulatory functions in multiple biological events such as cell cycle, differentiation, proliferation, apoptosis, metastasis, and invasion in various tumor cells [15,16]. Many researchers have explored the lncRNA expression profiling of CRC and found they can serve as biomarkers of CRC prognosis and diagnosis [17,18]. Of note, lncRNAs, extensively modified by m6A, can change the transcript stability and gene expression, bringing about regulatory abnormalities, which in turn influence the tumorigenesis and progression of cancer [5]. For example, METTL3-mediated m6A modification leads to lncRNA ABHD11-AS1 upregulation through stabilizing its RNA transcript, thereby promoting the Warburg effect of lung cancer [19]. METTL14 has been shown to suppress proliferation and metastasis of CRC via inhibiting lncRNA XIST expression [20]. However, m6A-modified lncRNAs in CRC are not fully understood.

In the present study, we investigated the correlations of m6A-related lncRNAs with clinicopathological features OS, programmed death-ligand 1 (PD-L1), and tumor immune microenvironment in CRC. We performed clustering subtypes and am6A-modified lncRNA signature which can accurately assess prognostic risk stratification for patients with CRC. We further explored the relationship between m6A-modified lncRNA signature and clinicopathological features and OS, PD-L1, immune scores, tumor-infiltrating immune cells. Furthermore, an accurate nomogram was established based on the lncRNA signature and clinicopathological variables to predict OS for CRC patients. In total, the current investigation might make great progress on the exploration of prognostic m6A-modified lncRNAs and shed new insights on the possible mechanism of CRC development.

## 2. Materials and Methods

### 2.1. Obtaining and Preprocessing Data

The transcriptome and clinical data of 571 CRC patients were downloaded from The Cancer Genome Atlas (TCGA; https://ocg.cancer.gov/programs/TCGA accessed on 12 February 2021) database, including mRNA and lncRNA expression data. After screening, patients with no clinical information were excluded. A total of 509 cases with corresponding tumor tissues and clinical information were included in the analysis. All cases were randomly assigned into the training cohort (*n* = 256) for m6A-modified lncRNA signature construction and the validation cohort (*n* = 253) for model validation, respectively, to reach a more compelling and reliable conclusion.

### 2.2. Identification of m6A-Modified lncRNAs and Univariate Survival Analysis

According to previous publications [6,21], 23 recognized m6A regulators were obtained, including writers, erasers, and readers (Appendix A). The m6A-modified lncRNAs were identified by Pearson correlation analysis (|R| > 0.4 and *p* < 0.001). Approval from the Institutional Ethics Committee is not required for the current study, as the TCGA data is publicly available. Univariate Cox regression analysis was assessed using the R survival coxph function (*p* < 0.05) to identify the prognostic m6A-modified lncRNAs.

### 2.3. Identification of Molecular Subtypes Associated with Prognosis

The “ConsensusClusterPlus” package in R was used for consensus clustering to identify distinct molecular patterns based on the expression of m6A-modified lncRNAs obtained from univariate Cox regression analysis. Eighty percent of the samples were sampled 100 times using a resampling procedure. The similarity distance between samples was estimated by the Euclidean distance, and K-means was used for clustering. The criterion for determining the number of clusters is that the coefficient of variation is relatively low, the consistency within the cluster is relatively high, and the area under the cumulative distribution function (CDF) curve has not increased significantly.

### 2.4. Clinical Significance of Molecular Subtypes

To explore the clinical value of molecular subtypes in CRC, we investigated the relationship between molecular subtypes, clinical characteristics, and prognosis. The characteristics of patients include age, tumor location, tumor status, and tumor stage. Subsequently, the differences in overall survival (OS) among different clusters were calculated by employing the Kaplan–Meier method and visualized by using the “survival” and “survminer” modules in R software.

### 2.5. Immune Infiltrating Analysis

CIBERSORT (https://cibersort.stanford.edu/ accessed on 2 February 2021), a deconvolution algorithm conducted for the characterization of complex cell compositions according to the gene expression spectra, has elaborated the significance in the examination of RNA mixtures for cellular biomarkers and therapeutic targets. CIBERSORT algorithm was utilized to precisely measure the fractions of 22 human immune cell subsets in CRC samples [22]. The tumor immune cell microenvironment (TME) score of each single CRC patient was estimated using the ESTIMATE algorithm to validate the effect of ssGSEA immune clusterin [23].

### 2.6. Establishment and Evaluation of the m6A-Modified lncRNA Signature

Univariate Cox regression analysis was conducted in the training cohort to carried m6A-modified prognostic lncRNAs. *p* < 0.05 was set as cutoff criteria. With the help of the R package “glmnet”, the least absolute shrinkage and selection operator (LASSO) regression analysis was applied to obtain the optimal candidates and construct a prognostic signature.

### 2.7. The Following Formula Was Used:

*Risk**Score* = Σ(*Expi* ∗ *Coefi*)

Expi is the expression level of the lncRNA, and Coefi is the estimated regression coefficient of the lncRNA. All CRC patients were assigned to high- and low-risk groups based on the risk score. We performed the survival analysis between the two risk groups to detect whether the difference in OS was subsistent. The receiver operating characteristic (ROC) curves were drawn using the R package “survivalROC” to determine the prognostic performance of the survival model. To validate the predictive accuracy of the signature, we applied the survival model to the validation cohort and generated the Kaplan–Meier survival curve and ROC curves to validate its efficiency.

### 2.8. Stratification Analysis of the m6A-Modified lncRNA Signature

The stratified analysis was used to assess whether the signature retains its predictive ability in various subgroups. These variables include age (<60 and >60 years), gender (female and male), tumor stage (I–II and III–IV), grade (G1–2, G3), T stage (T1–2 and T3–4), N stage (N0 and N1–2), M stage (M0 and M1), tumor location (left-side and right-side), lymphatic invasion (yes and no), pretreatment CEA level (>5 and <5), residual tumor (R0 and R1–2), and venous invasion (yes and no). In addition, to explore the impact of the signature on the clinicopathological characters in CRC, the correlation between the signature and clinicopathological variables of CRC was assessed using the Chi-square test and visualized using the “pheatmap” package and “ggpubr” in R software.

### 2.9. Gene Set Enrichment Analysis (GSEA)

To explore the illustrate the difference in survival among different risk score groups, the genome-wide expression profiles of the CRC patients were subjected to gene set enrichment analysis (GSEA, v4.0.3; http://www.broadinstitute.org/gsea accessed on 5 March 2021) to determine the genes that were differentially expressed between the high- and low-risk group patients. As the classical gene set in Molecular Signatures Database (MSigDB, https://www.gsea-msigdb.org/gsea/msigdb accessed on 5 March 2021), “c2.cp.kegg.v7.0.symbols.gmt (Curated)” was considered. One thousand random sample permutations and enriched gene sets with a nominal *p* < 0.05 were included in the analysis, and FDR < 0.25 were considered as the significant difference criteria. The rest of the parameters were set to the default values.

### 2.10. Development and Evaluation of a Prognostic Nomogram

To quantitatively evaluate CRC prognosis in clinical practice, the risk score and other clinicopathological variables, such as age, sex, tumor location, and TNM stage, were included in the univariate and multivariate Cox regression analyses. A nomogram that integrated both the m6A-modified lncRNA signature and clinicopathological characteristics was built internally in the training cohort and validated in the validation cohort for predicting 3- and 5-year OS. The predictive ability of the nomogram was evaluated by the calibration curves and the ROC curve for 3-year and 5-year using the risk score calculated by the m6A-modified lncRNA signature.

### 2.11. RNA Isolation and RT-qPCR

We collected six pairs of colorectal cancer tissues and adjacent normal tissues from patients who recently underwent surgical treatments in the Department of Gastrointestinal Surgery, Renmin Hospital of the University of Wuhan. Fresh tissues were frozen and stored at −80 °C. This research was approved by the Medical Ethics Committee of Renmin Hospital of the University of Wuhan. Informed consent was acquired from each involved patient. Total RNA from tissues of CRC patients was extracted using TRIzol reagent (Invitrogen; Carlsbad, CA, USA) according to the manufacturer’s protocol. For complementary DNA (cDNA) synthesis, 1 μg of total RNA and the PrimeScript RT reagent kit (Takara, Otsu, Shiga, Japan) were utilized. The SYBR-Green assay (Takara) was used to perform RT-qPCR and the progression was executed on a CFX-96 system (Bio-Rad Laboratories, Inc., Hercules, CA, USA). The GAPDH was used as an internal reference, and the relative lncRNA expression was calculated using the 2−ΔΔCq method. Primer sequences for qRT-PCR used in this study are shown in Appendix A.

### 2.12. Statistical Analyses

Immune infiltration analysis was investigated using the ‘gsva’ package in R. The Kaplan–Meier plot was used to generate survival curves and the Log-rank test was performed to evaluate the statistically significant differences. Univariate and multivariate Cox proportional hazard regression analyses were utilized to determine whether the m6A-modified lncRNA signature could be an independent prognostic factor. The nomogram was created by the “Survival” and “RMS” packages of R 3.6.3 could be used to provide a visual risk prediction. All statistical analyses were implemented using R version 4.0.4. Statistical significance was set at *p* < 0.05.

## 3. Results

### 3.1. Identification of m6A-Modified lncRNAs in CRC

The entire analytical process of the study is presented in Figure 1. We analyzed the transcriptome of 571 CRC samples and 44 normal samples from the TCGA database and identified 14,087 lncRNAs. Then, the expression matrixes 23 recognized m6A regulators from previous publications that were extracted from the TCGA. We used Pearson correlation analysis to screen m6A-modified lncRNAs (|R| > 0.4 and *p* < 0.001), and identified 152 lncRNAs that were significantly associated with m6A regulators. To consider whether the 152 m6A-modified lncRNAs were closely associated with the OS of CRC samples, we performed univariate Cox regression analysis to identify the prognostic value of the m6A-modified lncRNAs. Twenty-nine m6A-modified lncRNAs related to the OS time (*p* < 0.05) were screened out and applied in the following analysis (Appendix A).

### 3.2. Consensus Clustering Identified Distinct Molecular Subtypes

To identify the molecular subtypes, we performed the consistent clustering of 29 m6A-modified prognostic lncRNAs. The optimal number of clusters was determined according to comprehensive consideration of CDF and clinical significance. We choose the value of *k* = 2 as the appropriate number of clusters for further analysis (Figure 2A,B). Finally, 509 CRC samples were classified into two subgroups, namely, cluster1 (*n* = 126) and cluster 2 (*n* = 383) (Figure 2A–C).

### 3.3. Correlation of Molecular Subtypes Associated with Characteristics and Survival

Compared with cluster 1, the expression of m6A-modified lncRNA is lower in cluster 2, such as RPARP-AS1, AC010463.3, AC016394.3, and AP006621.2 (Figure 2D). To further examine the clinicopathological characteristics of the two subgroups identified by consensus clustering, the clinicopathological features were compared in different clusters of patients. The result demonstrated that Cluster 1 was preferentially related to advanced tumor stage (*p* < 0.05; Figure 2D) and with tumor status (*p* < 0.05; Figure 2D). In addition, we analyzed the difference in OS rate between the two subgroups. Patients with cluster 1 have a worse OS than those with Cluster 2 (Figure 2E). Taken together, clustering subgroups are significantly correlated with the heterogeneity of CRC.

### 3.4. Correlation of m6A-Modified lncRNA with PD-L1 and Tumor Immune Cell Microenvironment in CRC

To investigate the relationship between PD-L1 and m6A-modified lncRNA, we assessed the expression differences between cluster 1 and cluster 2 and the relationship between PD-L1 and m6A-modified lncRNA. Compared with normal colorectal tissues, PD-L1 expression was elevated in CRC tissues (*p* < 0.05; Figure 3A). A lower PD-L1 expression in cluster1 was observed than that in cluster 2 (*p* < 0.05; Figure 3B). The PD-L1 was negatively correlated with the expression levels of AL391684.1, AL138756.1, AC107375.1, ZKSCAN2-DT, AL161729.4, AC016737.1, AC013652.1, and AC005229.4, and was significantly positively correlated with AC245041.1 expression level (Figure 3C).

To investigate the role of m6A-modified lncRNAs on the TME of CRC, we evaluated stromal scores, immune scores, and estimate scores of two subtypes by the ESTIMATE package, and calculated the proportion of 22 immune cell types by CIBERSORT. The results demonstrated that the stromal score, immune score, and ESTIMATE score were lower in cluster 1 (Figure 3D–F). Cluster 1 had a higher infiltration level of CD4 memory-activated T cells, T cells CD4 naïve, and eosinophils, while high NK cells activated, B cells memory, T cells regulatory, monocytes, and macrophages M2 levels were observed in cluster 2 (Figure 3G).

### 3.5. Development and Validation of m6A-Modified lncRNA Prognostic Signature

To further explore the prognostic value of m6A-modified lncRNAs in CRC, we divided 509 CRC patients into the training cohort (*n* = 256) and validation cohort (*n* = 253), in a random manner at a ratio of 1:1. The result showed that the groupings were reasonable and all comparisons between the training and validation cohort were not significantly different by the chi-square test (*p* > 0.05; Table 1).

To reduce the overfitting among prognostic markers, 29 m6A-modified prognostic lncRNAs were further analyzed with LASSO Cox analysis, and 7 lncRNAs (AC003101.2, ITGB1-DT, AC245041.1, ZKSCAN2-DT, AL391422.4, AC156455.1, and AC005229.4), were filtered out in the training cohort (Figure 4A,B). Then, a seven-gene m6A-modified prognostic signature was established using the adjusted regression coefficients of each gene, and the risk score was calculated as the expression level of AC003101.2*0.594 + expression level of ITGB1-DT*0.141 + expression level of AC245041.1*0.142 + expression level of ZKSCAN2-DT*0.035 + expression level of AL391422.4*0.387 + expression level of AC156455.1*0.149 + expression level of AC005229.4*0.249. After scoring each patient’s risk through the signature, we divided the CRC samples into low- and high-risk groups according to the median risk score (1.905). The risk score of each patient was calculated and ranked on the basis of the signature model (Figure 4C). The scatter plot represented the OS status of CRC patients according to the risk score, and it was demonstrated that the higher the risk scores, the higher the number of death (Figure 4D). The expression profiles of the signature genes indicated that tumors with higher risk scores tended to exhibit elevated expression levels of the seven lncRNAs (Figure 4E). Kaplan–Meier plot revealed that patients in the high-risk group had significantly worse OS than those in the low-risk group in the training cohort (*p* < 0.05; Figure 4F). ROC curve analysis indicated that the survival model applied to the training cohort had a good predictive value, and the AUC of the ROC curves was 0.704 (Figure 4G). To confirm that the m6A-modified signature had a robust prognostic value, the same analysis was performed using the validation cohort, and the results were consistent with those derived from the training cohort (Figure 5A–E). Similarly, the difference in the Kaplan–Meier survival curve between high-risk and low-risk groups is statistically significant (Figure 5D) and the AUC values of ROC curve analyses were 0.695, at 3 years (Figure 5E).

### 3.6. Stratification Analysis of the m6A-Modified lncRNA Signature

The stratified analysis was used to assess whether the signature retains its predictive ability in various subgroups. The results demonstrated that the age, gender, tumor stage, tumor location, pretreatment CEA level, the lymphatic and venous invasion could be divided into two groups with significant prognostic differences based on the signature (*p* < 0.05; Figure 6A–N). Moreover, the differences in risk scores were assessed by various clinicopathological variables (Figure 7). The risk scores showed significant statistical differences in clustering subtypes (*p* < 0.05), immune score (*p* < 0.05), tumor stage (*p* < 0.05), lymphatic and venous invasion (*p* < 0.05). Specifically, cluster 1 showed statistically significantly lower risk scores (*p* < 0.05; Figure 7A). Patients with advanced tumor stage tend to have distinctly higher risk scores than those with early-stage (*p* < 0.05; Figure 7B). Patients with high immune scores had significantly lower risk scores than patients with low immune scores (*p* < 0.05; Figure 7C). Patients with lymphatic and venous invasion had significantly higher risk scores than patients without lymphatic and venous invasion (*p* < 0.05; Figure 7D,E).

### 3.7. Gene Set Enrichment Analysis (GSEA)

To elucidate the biological functions of the seven m6A-modified lncRNA signature, GSEA was further used to investigate the key signatures of different risk groups. The results revealed that ECM-receptor interaction, focal adhesion, Hedgehog signaling pathway, were enriched in the high-risk group (Appendix A). These findings indicated that m6A-lncRNAs may participate in the progression of CRC via the above pathways, which partially explained the poor survival in the high score group and may help us gain insight into the implication of m6A-modified lncRNA signature.

### 3.8. Validation of the Expression Levels of Seven lncRNAs of Prognostic Signature

To further verify the accuracy of the m6A-modified lncRNA signature, expression levels of seven m6A-modified prognostic lncRNAs were measured in six colorectal cancer tissues and six adjacent normal tissues using RT-qPCR. As shown in Appendix A, AC003101.2, ITGB1-DT, AL391422.4, AC156455.1, and AC005229.4 were upregulated in colorectal cancer tissues compared with corresponding normal tissues. Meanwhile, AC245041.1 and ZKSCAN2-DT were downregulated in colorectal cancer tissues compared with corresponding normal tissues.

### 3.9. Development of a Nomogram for Prognosis Prediction

To detect whether the prognostic value of this signature is depended on the clinicopathological parameters, univariate and multivariate Cox analyses were adopted to analyze the following variables such as risk score, age, gender, tumor location, and tumor stage. The results demonstrated that the m6A-modified lncRNA signature is an independent clinical prognostic factor for patients with CRC in the training cohort by multivariate Cox analysis (HR = 1.374, 95% CI = 1.206–1.566, *p* < 0.05; Figure 8A,B). To provide the surgeons with a quantitative method for prognosis prediction of CRC patients, we built a nomogram to predict 3-year and 5-year OS of patients by integrating tumor stage, tumor location, and the risk score in the training set (Figure 8C). With the help of the nomogram, the prognosis can be effectively predicted based on the individual characteristics of the patient. The AUCs of the nomogram for predicting 3- and 5-year OS were 0.819 and 0.826 in the training cohort (Figure 8D) and 0.757 and 0.776 in the validation cohort (Figure 8E). To compare the consistency of the model prediction with actual clinical outcomes, calibration curves for 3- and 5-year OS in the training (Figure 8F,G) and validation cohorts (Figure 8H,I) were constructed. The results for predicting 3- and 5-year OS showed strong consistency between the nomogram prediction and the actual observation (Figure 8F–I).

## 4. Discussion

CRC is a heterogeneous and highly malignant tumor with high morbidity and mortality [24]. Biomarkers have crucial functions in the diagnosis and treatment of CRC patients. For cancer patients with early-stage disease, biomarkers can be used as a detection method to identify susceptibility or early-stage disease. For patients with advanced cancer, biomarkers can be used as therapeutic targets to improve the survival time of patients. Therefore, exploring the molecular mechanisms and screening reliable prognostic biomarkers are urgently needed to accurately predict clinical outcomes of CRC patients and provide indispensable information to guide personalized treatment.

The m6A modification plays a vital role in post-transcriptional regulation of gene expression [25] and has been discovered in diverse non-coding RNAs and approximately 67% lncRNAs of 3′ UTRs exist m6A peaks [5]. Accumulating evidence indicates that the m6A modifications of lncRNA may affect the development and progression of cancers. The m6A regulators reportedly act as a lncRNA structural switch, participates in the lncRNA-mediated competing endogenous RNA model, and enhances the stability of lncRNA to serve its functions, thereby influencing tumor initiation and progression [26,27,28]. For example, m6A-induced LNCAROD can promote the development of head and neck squamous cell cancer by forming a ternary complex with YBX1 and HSPA1A [26]. Therefore, m6A modification is targeted at lncRNA, and more attention should be paid to the interaction and function of lncRNA and m6A modification to investigate efficient prognostic biomarkers or provide patients with an optimal therapeutic approach. However, how it acts in a lncRNA-dependent manner in the progression of CRC is still unknown. Moreover, it is necessary to further study m6A-modified lncRNAs to clarify the potential regulatory mechanism of m6A-modified lncRNAs in the tumor immune microenvironment.

A growing body of evidence has shown that epigenetic regulators are essential for the reversal of the process of misregulation [29]. An RNA methylation-targeted system using a Cas13-directed methyltransferase has potential and has been used with encouraging results [30]. In recent years, a set of specific or non-specific small molecule m6A inhibitors have shown strong anti-tumor effects in many types of cancer [31,32]. Yankova et al. [32] revealed that STM2457 is a highly effective and selective METTLE3 inhibitor, which can reduce the growth of acute myeloid leukemia (AML) and increase differentiation and apoptosis. AML mice treated with STM2457 prolonged their survival. Similarly, two small-molecule FTO inhibitors (CS1 and CS2) have been shown to significantly attenuate the self-renewal and reprogram immune response of leukemia stem/starter cells by suppressing the expression of immune checkpoint genes [31]. These studies highlight the broad potential of targeting m6A for cancer therapy.

In the present study, we identified seven m6A-modified lncRNAs by Pearson’s correlation analysis between 23 m6A regulators and lncRNAs in 509 CRC patients from the TCGA database and identified 29 m6A-modified prognostic lncRNAs. We successfully divided CRC into two molecular subtypes, and the two cluster subtypes exhibited significantly different survival and clinical characters. Furthermore, the two subtypes were significantly associated with PD-L1, immune score, and immune cell infiltration levels. The significant difference in survival between the two subtypes may be linked to the higher immune score and PD-L1 in cluster 2, which is consistent with the results of previous studies [33]. Taken together, m6A-modified lncRNAs might serve as a biomarker for prognosis prediction of CRC patients. Therefore, we performed LASSO Cox regression analysis and constructed a seven m6A-modified lncRNA prognostic signature. A high-risk score was associated with worse OS and advanced clinicopathological characteristics. The predictability of the signature was confirmed by the tdROC curve and the validation cohort. Patients in the two risk groups exhibited significantly different distinct molecular patterns, PD-L1 expression, and immune score. Moreover, a stratified analysis indicated that the signature retains its predictive ability in the various subgroups. Multivariate analysis indicated that the signature was also an independent predictor compared with other clinicopathological characteristics. To provide the physicians with a quantitative method for prognosis prediction of CRC patients, we integrated the signature with clinicopathological variables to establish a robust nomogram with good reproducibility and reliability. The m6A-modified lncRNA signature can be used for prognosis stratification of CRC patients and will assist with understanding the molecular mechanism of CRC and provide new ideas for targeting therapies.

The proposed signature contained seven m6A-modified lncRNAs. Among the seven lncRNAs, AL391684.1, AL138756.1, AC107375.1, AL161729.4, AC016737.1, and AC013652.1 had not been previously reported. ZKSCAN2-DT, also known as CTD-2547G23.4, was overexpressed in hepatocellular carcinoma (HCC) cell lines and clinical specimens and was associated with tumor grade and vascular tumor cell type [34]. Depletion of ZKSCAN2-DT in the Huh-7 cell line can arrest the cell cycle of HCC cells. Several studies indicated that AC005229.4 was associated with autophagy-related lncRNAs by Pearson’s correlation analysis [35,36]. Robust evidence has shown that there is an intimate relationship between m6A modification and autophagy in cancer [37,38]. For example, m6A reader YTHDF1 can drive hypoxia-induced autophagy and malignancy of HCC by promoting ATG2A and ATG14 translation [37]. Liu et al. indicated that METTL3-mediated autophagy can reverse the gefitinib resistance of NSCLC cells by β-elemene [39]. LncRNA AC005229.4 may affect the autophagy of CRC by regulating m6A-modified lncRNA, although further experiments are required to verify this speculation.

Nevertheless, a few limitations of this present study should be taken into consideration. First, this study was not comprehensive enough, it is necessary to perform more molecular biology experiments focused on the mechanism of m6A-modified lncRNAs in the progression of CRC. Second, although the m6A-modified lncRNA signature and nomogram were validated in the TCGA validation cohort, larger sample size CRC cohorts from different technology platforms should also be used to provide further validation. Third, there is no real experimental validation of the m6A modifications in CRC for the seven lncRNAs in the nomogram.

In conclusion, we systematically assessed prognostic significance, the association of PD-L1, its role in the tumor immune microenvironment, and potential regulatory mechanisms of m6A-modified lncRNAs in CRC. A seven m6A-modified lncRNA prognostic signature was established in our present study, which might act as an independent prognostic variable for patients with CRC. Moreover, we established a prognostic nomogram incorporating the gene signature, tumor location, and tumor stage to predict the OS of patients with CRC. The novel model might provide insights for predicting the prognosis of CRC patients and suggestions to guide individual therapeutic strategies.

## Figures and Tables

**Figure 1 genes-12-01375-f001:**
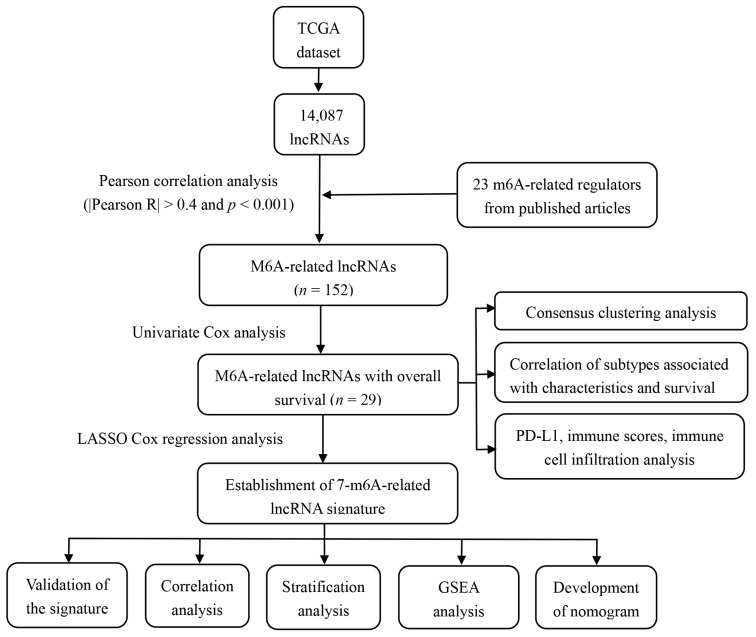
The entire analytical process of the study.

**Figure 2 genes-12-01375-f002:**
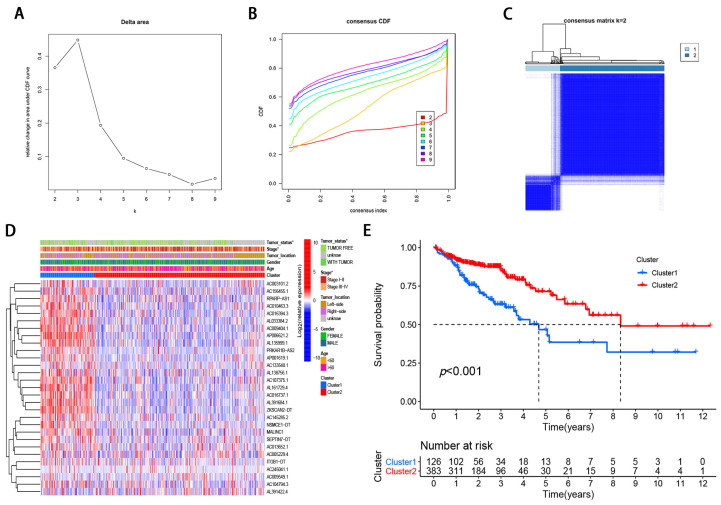
Differential clinicopathological features and survival of CRC in two cluster subtypes. (**A**) Consensus clustering cumulative distribution function (CDF) for k = 2 to 9. (**B**) Relative change in area under CDF curve for k = 2–9. (**C**) Consensus clustering matrix for k = 2. (**D**) Heatmap of correlation of the two clusters with clinicopathologic features. (**E**) Kaplan–Meier curves of overall survival for patients with CRC in two clusters.

**Figure 3 genes-12-01375-f003:**
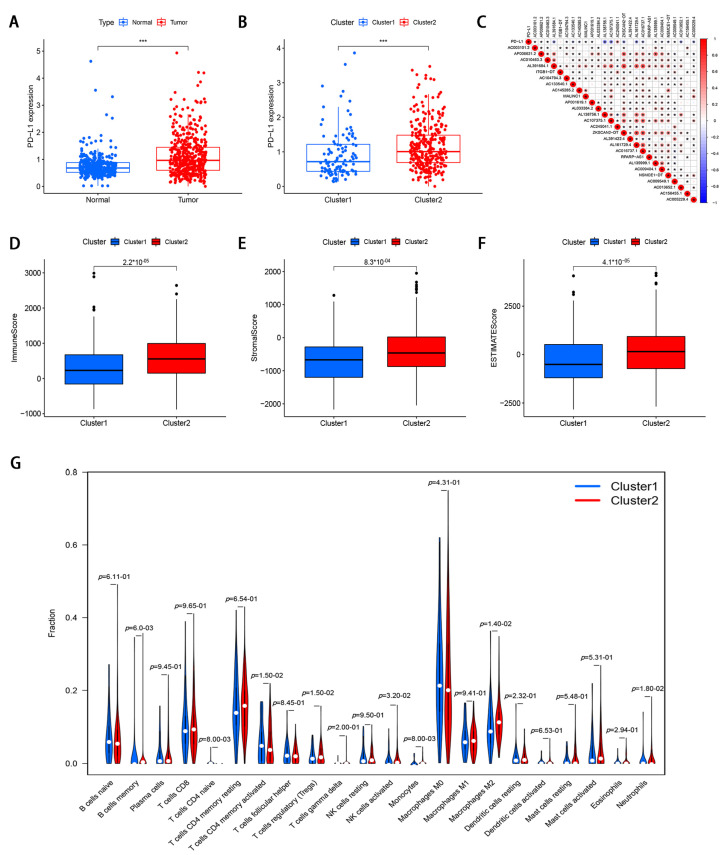
The correlation of m6A-modified lncRNA with PD-L1 and tumor immune cell microenvironment in CRC. (**A**) The expression level of PD-L1 in CRC tissues and normal adjacent tissues. (**B**) The expression level of PD-L1 in cluster 1 and cluster 2 (*** *p* < 0.001). (**C**) The correlation of PD-L1 with m6A-modified lncRNA. (**D**–**F**) immune score, stromal score, and ESTIMATE score in cluster 1 and cluster 2. (**G**) The infiltrating levels of 22 immune cell types in cluster 1 and cluster 2.

**Figure 4 genes-12-01375-f004:**
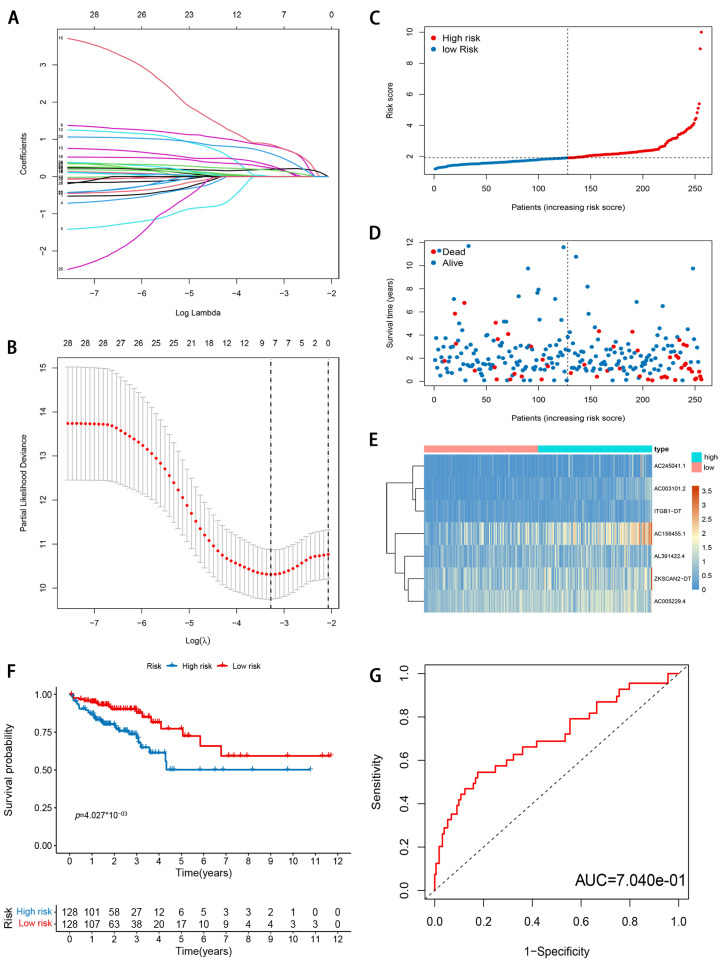
Development and evaluation of m6A-modified lncRNA signature related to OS in TCGA training cohort. (**A**) LASSO coefficient profiles of seven prognosis-associated lncRNAs. (**B**) Tenfold cross-validation for tuning parameter selection in the LASSO model. (**C**) The risk score distribution. Blue dots represent risk score for low-risk patients; red dots represent risk score for high-risk patients. (**D**) The relationship between survival status and risk score. The abscissa represents the number of patients, and the ordinate is the risk score. Red dots represent dead patients, blue dots are living patients. (**E**) Heatmap showing the expression profile of the seven signature lncRNAs. (**F**) Kaplan–Meier method was used to plot the RFS curve for the high-risk score and low-risk score groups. (**G**) ROC curves and AUCs of the 7-gene signature.

**Figure 5 genes-12-01375-f005:**
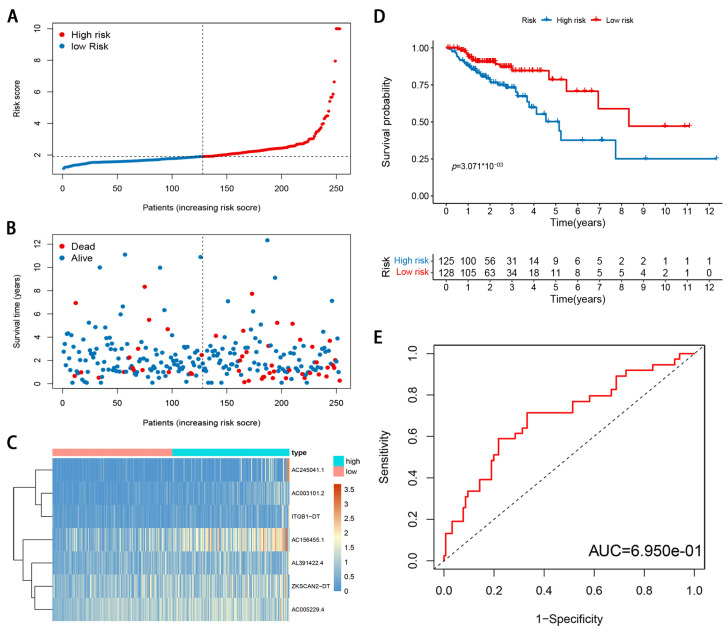
Validation of m6A-modified lncRNA signature in TCGA validation cohort. (**A**) The risk score distribution. Blue dots represent risk score for low-risk patients; red dots represent risk score for high-risk patients. (**B**) The relationship between survival status and risk score. The abscissa represents the number of patients, and the ordinate is the risk score. Red dots represent dead patients, blue dots are living patients. (**C**) Heatmap showing the expression profile of the seven signature lncRNAs. (**D**) Kaplan–Meier method was used to plot the RFS curve for the high-risk score and low-risk score groups. (**E**) ROC curves and AUCs of the 7-gene signature.

**Figure 6 genes-12-01375-f006:**
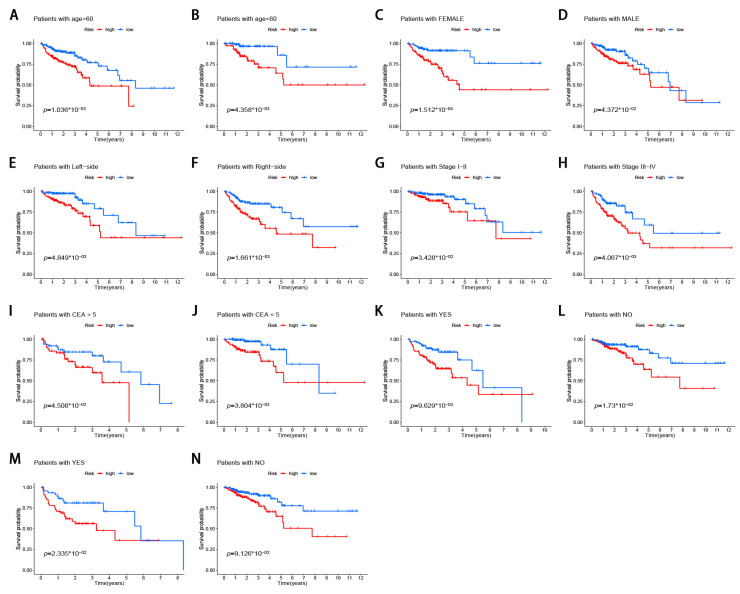
Stratification analysis of the m6A-modified lncRNA signature in CRC. (**A**,**B**) age (<60 or >60 years old). (**C**,**D**) gender (female or male). (**E**,**F**) tumor location (left-side or right-side). (**G**,**H**) tumor stage (I–II or III–IV). (**I**,**J**) pretreatment CEA level (<5 or >5). (**K**,**L**) lymphatic invasion (yes or no). (**M**,**N**) venous invasion (yes or no).

**Figure 7 genes-12-01375-f007:**
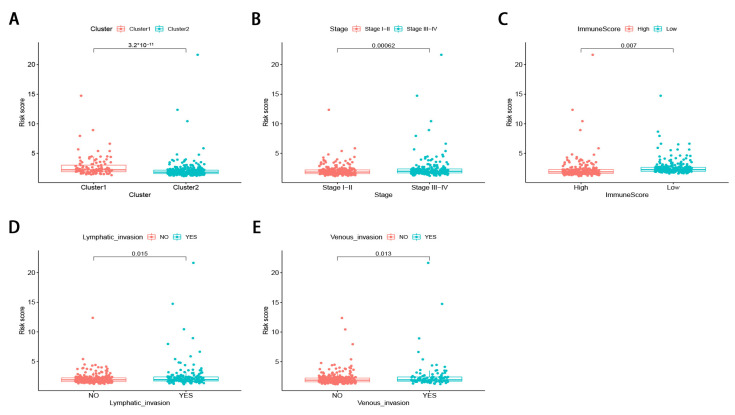
The differences in risk scores by various clinicopathological variables in CRC. (**A**) cluster 1 showed statistically significantly lower risk scores. (**B**) Patients with advanced tumor stage tend to have distinctly higher risk scores. (**C**) patients with high immune score had significantly lower risk scores. (**D**,**E**) patients with lymphatic and venous invasion had significantly higher risk scores.

**Figure 8 genes-12-01375-f008:**
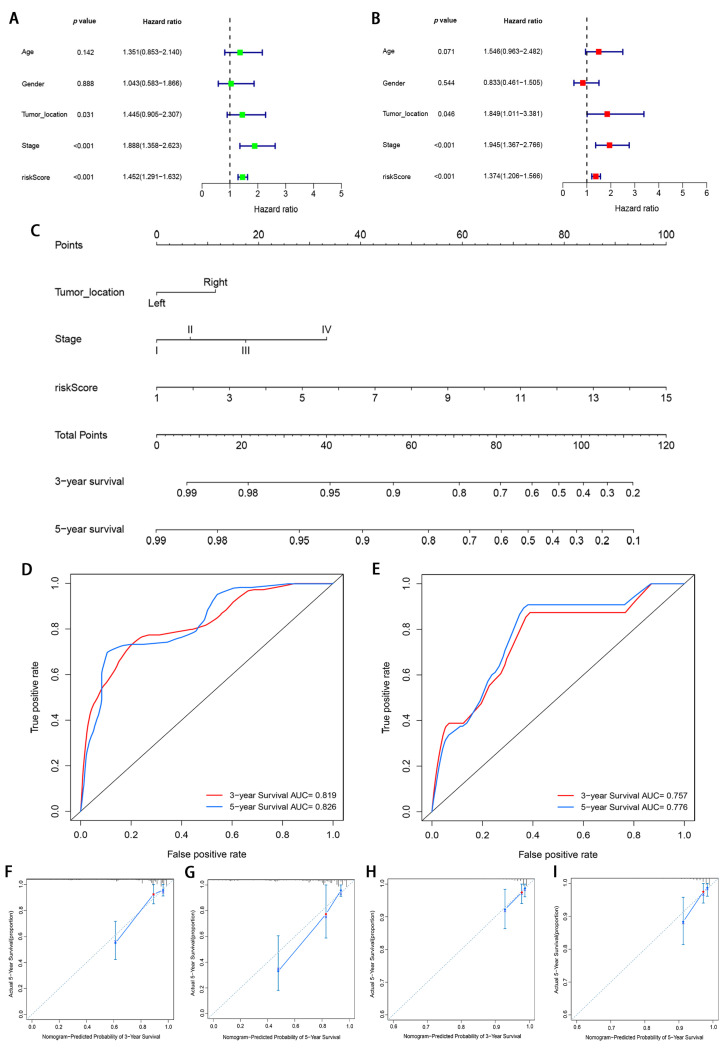
Construction and validation of a nomogram. (**A**,**B**) Forest plots of univariate and multivariate Cox regression analysis of variables associated with OS in the training cohort; (**C**) The nomogram for predicting the 3- and 5-year OS of CRC patients in the training cohort. (**D**,**E**) ROC curves for predicting the 3- and 5-year ROC curves in the training cohort (**D**) and validation cohort (**E**). (**F**–**I**) Calibration curves of the nomogram for predicting of 3- and 5-year OS in the training cohort (**F**,**G**) and validation cohort (**H**,**I**).

**Table 1 genes-12-01375-t001:** Clinicopathologic characteristics of colorectal cancer patients in the TCGA cohort.

Variable	Total	TCGA Training Cohort	TCGA Validation Cohort	*p*-Value
	(*n* = 509)	(*n* = 256)	(*n* = 253)	
	N (%)	N (%)	N (%)	
Age				0.566
≤60	163 (32.0)	85 (33.2)	78 (30.8)	
>60	346 (68.0)	171 (66.8)	175 (69.2)	
Gender				0.903
Female	232 (45.6)	116 (45.3)	116 (45.8)	
Male	277 (54.4)	140 (54.7)	137 (54.2)	
Tumor location				0.593
Left	251 (49.3)	132 (51.6)	119 (47.0)	
Right	237 (46.6)	114 (44.5)	123 (48.6)	
Unknow	21 (4.10	10 (3.9)	11 (4.3)	
Stage				0.878
I–II	278 (54.6)	140 (54.7)	138 (54.5)	
III–IV	213 (41.8)	108 (42.2)	105 (41.5)	
Unknown	18 (3.5)	8 (3.1)	10 (4.0)	
Pretreatment CEA level				0.737
<5	212 (41.7)	110 (43.0)	102 (40.3)	
>5	112 (22.0)	53 (20.7)	59 (23.3)	
Unknown	185 (36.3)	93 (36.3)	92 (36.4)	
Tumor status				0.759
Tumor free	155 (30.5)	75 (29.3)	80 (31.6)	
With tumor	58 (11.4)	28 (10.9)	30 (11.9)	
Unknown	296 (58.2)	153 (59.8)	143 (56.5)	
Venous invasion				0.845
No	324 (63.7)	166 (64.8)	158 (62.5)	
Yes	110 (21.6)	54 (21.1)	56 (22.1)	
Unknown	75 (14.7)	36 (14.1)	39 (15.4)	
Lymphatic invasion				0.223
No	274 (53.8)	147 (57.4)	127 (50.2)	
Yes	177 (34.8)	84 (32.8)	93 (36.8)	
Unknown	58 (11.4)	25 (9.8)	33 (13.0)	

## Data Availability

The datasets analyzed for this study can be found in the TCGA database (http://www.cancer.gov/tcga accessed on 12 February 2021).

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
