# Peer review of "N6-Methyladenosine-Related lncRNA Signature Predicts the Overall Survival of Colorectal Cancer Patients"

_genes, 2021, doi:10.3390/genes12091375_

Round 1

Reviewer 1 Report

In this paper the authors assess the prognostic value of m6A-related lncRNAs in CRC patients. Specifically, they correlate m6A-related lncRNA signature with OS in CRC patients.

The paper is interesting, however, in my opinion some concerns should be addressed.

1) The valdidation of lncRNA expression and the analysis of  m6A in relation with CRC progression should be addressed.

2) In the light of the growing interest on the role of m6A pharmacological inhibition as a new strategy for tumor growth and development, a discussion on this topic should be added (for refs. see also Yankova 2021 and Garbo 2021).

Author Response

1.The validation of lncRNA expression and the analysis of m6A in relation with CRC progression should be addressed.

Answer: Expression levels of 7 m6A-related prognostic lncRNAs were measured in 6 colorectal cancer tissues and 6 adjacent normal tissues using RT-qPCR (Supplementary Figure 2). In addition, we have investigated the correlation between m6A signature and clinicopathological features and found that the signature was associated with advanced tumor stage, lymphatic and venous invasion.

2.In the light of the growing interest on the role of m6A pharmacological inhibition as a new strategy for tumor growth and development, a discussion on this topic should be added (for refs. see also Yankova 2021 and Garbo 2021).

Answer: A discuss on m6A pharmacological inhibition has been added and the two articles have been cited in the revised version.

Reviewer 2 Report

I found this manuscript interesting. Similar studies have been undertaken recently by the scientists also for other types of cancer. Results of presented study have potential for use in future for determining the prognosis in clinical practice, however I have some comments for the authors as detailed below.

  • Introduction - the second sentence repeats the content of the first sentence.
  • Results - line 182 - according to the earlier information (line 94) - it should probably be 571 instead of 577.
  • Figure 1 - In the analytical process diagram, the sub-step entitled "Consensus clustering analysis" should probably be placed a little lower in order to be consistent with the description in the text.
  • Line 195 – mistake: instead of Figure 3 it should be Figure 2
  • Inscriptions are not legible on some charts
  • Lines 313-316- it is worth adding relevant references
  • Line 335 - the word "lncRNAs" was lost in the sentence: “…seven m6A-related prognostic signature.”
  • Line 494 - description of Figure 6 -mistake: the description is inconsistent with the order of the charts placed – E-F – instead of “tumor stage” it should be “tumor location”; G-H - instead of “tumor location” it should be “tumor stage”
  • Line 498 - description of Figure 7D - remove "venous invasion" from the description of plot "D".
  • Figures 5A and D, Figures 4Cand F - It would be good to use the same color consistently in the charts for high risk and low risk
  • The manuscript should be revised in terms of some syntactical/grammatical/typographical

Author Response

1.Introduction - the second sentence repeats the content of the first sentence.

Answer: The repeated sentence has been deleted.

2.Results - line 182 - according to the earlier information (line 94) - it should probably be 571 instead of 577.

Answer: It should be 571 and correction has been made.

3.Figure 1 - In the analytical process diagram, the sub-step entitled "Consensus clustering analysis" should probably be placed a little lower in order to be consistent with the description in the text.

Answer: The sub-step entitled "Consensus clustering analysis" has been placed in the right location.

4.Line 195 – mistake: instead of Figure 3 it should be Figure 2.

Answer: Correction has been made in revised version.

5.Inscriptions are not legible on some charts.

Answer: The font of several Figures has been adjusted.

6.Lines 313-316- it is worth adding relevant references.

Answer: The relevant references have been added.

7.Line 335 - the word "lncRNAs" was lost in the sentence: “…seven m6A-related prognostic signature.”

Answer: Correction has been made.

8.Line 494 - description of Figure 6 -mistake: the description is inconsistent with the order of the charts placed – E-F – instead of “tumor stage” it should be “tumor location”; G-H - instead of “tumor location” it should be “tumor stage”.

Answer: Correction has been made in revised version.

9.Line 498 - description of Figure 7D - remove "venous invasion" from the description of plot "D".

Answer: Correction has been made.

10.Figures 5A and D, Figures 4Cand F - It would be good to use the same color consistently in the charts for high risk and low risk.

Answer: The colors of Figure 4F and Figure 5D have been changed.

11.The manuscript should be revised in terms of some syntactical/grammatical/typographical.

Answer: Syntactical/grammatical/typographical have been revised.

Reviewer 3 Report

Song, Ren et al. identified a lncRNA signature and established a prognostic nomogram to predict overall survival in colorectal cancer.

While the analyses they included are potentially very relevant, I am afraid some conclusions were not properly derived and their key threshold of |R| > 0.4 is very lenient to develop such a signature with critical future implications. Therefore, in its current form, I have some concerns which I think should be taken into consideration and commented in the manuscript:

Primary comments:

  • |R| > 0.4 is a very conservative value. I urge the authors to repeat the analyses with a much higher threshold (>= 0.7). Furthermore, other than correlation, there’s no real experimental validation of the m6A modifications in CRC for the lncRNAs they ended up keeping in the nomogram. Either this is investigated or included as a shortcoming of this study.
  • Please include the “23 recognized m6A regulators” as a table with appropriate references and a brief description.
  • Why do the authors perform a stratified analysis across subgroups and variables if two paragraphs later they mentioned they performed a multivariate approach with those variables as covariates?
  • Regarding GSEA, please include version of the software and all the different parameters used (scheme, number of permutations, which statistic was used to rank the features, ...). Furthermore, KEGG only contains protein coding genes in their gene sets, so authors should explain how they perform enrichment for such gene sets using lncRNAs as input.
  • “k = 2 as the appropriate… (Figure 3a and b)”. This figure label is wrong, it should be Figure 2A and B. Furthermore, it seems like k = 3 should have been a more appropriate number of clusters.
  • “The results 220 demonstrated that cluster 1 was negatively correlated with stromal score, immune score, 221 and ESTIMATE score (Figure 3D-F).”. That's a wrong interpretation of the output from the ESTIMATE score.
  • Regarding Figure 4C: What was the specific value (threshold) to split the data? Merely the half point or from maximally selected rank statistics?
  • On page 6 authors describe the risk score as a linear model with the expression of 7 lncRNAs. However, there was no intercept value included. Please explain why.
  • In my opinion, incorrect interpretation of Figure 7 (and incorrect figure legend). The p-value probably doesn't come from correlation but from difference in mean values (T-test). Statements such as "cluster 1 was significantly correlated with low-risk score" should be re-written as "cluster 1 showed statistically significantly lower risk scores".
  • In the statement “the m6A-related lncRNA signature is an independent clin-287 ical prognostic factor (P < 0.05) for patients with CRC in the training cohort (Figure 8A-B).”, please indicate which statistical test was used.
  • Regarding the 3- and 5-year OS analyses showed on Figures 8F-I, it's not clear to me for how many samples this 3- and 5-year OS information was available. It seems very strange that only 3 nanonagram-predicted probability values are possible (OX axes; 3 points with upper and lower whiskers). The Nanogram predicted probabilities should be continuous in the range of 0 to 1, instead of discrete.
  • Legend of Figure 2D: This seems incorrect: colour legend missing, but values from -10 to +10 cannot be correlation. To me it seems log2(expression).
  • Figure 3D-F: The depicted p-values come from which statistical test?
  • Please include a new version of Fig 4E where the rows are standardized (z-scores) because visually, it seems that only 2 out of 7 lncRNAs can really distinguish high vs low risk.

Minor comments:

- OS = overall survival. It should be defined the first time it appears on the manuscript (abstract).

- “Commonest” is not grammatically correct (should be “most common”). Please see https://dictionary.cambridge.org/es-LA/dictionary/english/common

- Please update the definition of lncRNA (not only length, but also based on presence of ORFs).

- “In present study” should be “In the present study”.

- "m6a-realted" should be "related". But nevertheless, I suggest the use of an alternative name throughout the manuscript: m6A-modified lncRNAs.

- “Then, we constructed clustering subtypes and m6A-related lncRNA signature”. "Then" should be removed. Clustering is performed, not constructed. It should be “and a m6A-related lncRNA signature which” (without comma).

- Before introducing CIBERSORT or ESTIMATE, please include a brief definition of deconvolution.

-“ (TME) score of each single CRC patient were estimated” should be “was estimated”.

Author Response

Primary comments:

1.|R| > 0.4 is a very conservative value. I urge the authors to repeat the analyses with a much higher threshold (>= 0.7). Furthermore, other than correlation, there’s no real experimental validation of the m6A modifications in CRC for the lncRNAs they ended up keeping in the nomogram. Either this is investigated or included as a shortcoming of this study.

Answer: It is well-known that |R| ≥ 0.4 can be considered as relevant, and many articles on lncRNA prognosis model use the correlation coefficient > 0.4 (or 0.3) as a threshold, including high score articles. For instance:

1). Immune-Related lncRNA to Construct Novel Signature and Predict the Immune Landscape of Human Hepatocellular Carcinoma;

2). Immune-related long noncoding RNA signature for predicting survival and immune checkpoint blockade in hepatocellular carcinoma;

3). Identification of Epithelial–Mesenchymal Transition-Related lncRNA With Prognosis and Molecular Subtypes in Clear Cell Renal Cell Carcinoma;

4). Identification of a prognostic ferroptosis-related lncRNA signature in the tumor microenvironment of lung adenocarcinoma.

In addition, expression levels of 7 m6A-related prognostic lncRNAs were measured in 6 colorectal cancer tissues and 6 adjacent normal tissues using RT-qPCR (Supplementary Figure 2). However, the m6A modifications in CRC for the 7 lncRNAs were not validated, which was listed as a shortcoming of this study.

2.Please include the “23 recognized m6A regulators” as a table with appropriate references and a brief description.

Answer: The 23 recognized m6A regulators have been provided as Supplementary Table 1 and related references have been cited.

3.Why do the authors perform a stratified analysis across subgroups and variables if two paragraphs later they mentioned they performed a multivariate approach with those variables as covariates?

Answer: Both stratified analysis and multivariate analysis are used to evaluate the prognostic performance of the signature. The stratified analysis is to assess whether the signature retains its predictive ability in various subgroups. The multiple Cox regression analysis is to evaluate whether the signature is an independent prognostic factor.

4.Regarding GSEA, please include version of the software and all the different parameters used (scheme, number of permutations, which statistic was used to rank the features, ...). Furthermore, KEGG only contains protein coding genes in their gene sets, so authors should explain how they perform enrichment for such gene sets using lncRNAs as input.

Answer: The version of the software and all the different parameters used have been provided. In addition, we used GSEA to analyze differentially expressed mRNAs between the high- and low-risk group patients.

5.“k = 2 as the appropriate… (Figure 3a and b)”. This figure label is wrong, it should be Figure 2A and B. Furthermore, it seems like k = 3 should have been a more appropriate number of clusters.

Answer: Correction has been made in the revised version. In addition, since the consistency within the cluster is relatively low, k = 2 was used to sort the entire cohort into two groups.

6.“The results 220 demonstrated that cluster 1 was negatively correlated with stromal score, immune score, 221 and ESTIMATE score (Figure 3D-F).”. That's a wrong interpretation of the output from the ESTIMATE score.

Answer: Correction has been made in revised version.

7.Regarding Figure 4C: What was the specific value (threshold) to split the data? Merely the half point or from maximally selected rank statistics?

Answer: We divided the CRC samples into high- and low-risk groups according to the median risk score (1.905).

8.On page 6 authors describe the risk score as a linear model with the expression of 7 lncRNAs. However, there was no intercept value included. Please explain why.

Answer: Intercept is the prediction when all expressions are zero. There is no need to include it in the model, in fact it should be deleted in some cases.

9.In my opinion, incorrect interpretation of Figure 7 (and incorrect figure legend). The p-value probably doesn't come from correlation but from difference in mean values (T-test). Statements such as "cluster 1 was significantly correlated with low-risk score" should be re-written as "cluster 1 showed statistically significantly lower risk scores".

Answer: This may be a generally accepted explanation, because many articles explain it this way. For instance:

  • Development and Validation of a 28-gene Hypoxia-related Prognostic Signature for Localized Prostate Cancer;
  • Immune-related long noncoding RNA signature for predicting survival and immune checkpoint blockade in hepatocellular carcinoma;
  • Identification of a prognostic immune signature for cervical cancer to predict survival and response to immune checkpoint inhibitors.

10.In the statement “the m6A-related lncRNA signature is an independent clin-287 ical prognostic factor (P < 0.05) for patients with CRC in the training cohort (Figure 8A-B).”, please indicate which statistical test was used.

Answer: The statistical test used has been added in revised version.

11.Regarding the 3- and 5-year OS analyses showed on Figures 8F-I, it's not clear to me for how many samples this 3- and 5-year OS information was available. It seems very strange that only 3 nomogram-predicted probability values are possible (OX axes; 3 points with upper and lower whiskers). The Nanogram predicted probabilities should be continuous in the range of 0 to 1, instead of discrete.

Answer: Nomogram-predicted probability values can be adjusted, but most studies show 3 or 4 points, including many articles published in J Clin Oncol. For instance:

1). Prognostic Nomogram for Intrahepatic Cholangiocarcinoma After Partial Hepatectomy;

2). Nomograms Predicting Progression-Free Survival, Overall Survival, and Pelvic Recurrence in Locally Advanced Cervical Cancer Developed From an Analysis of Identifiable Prognostic Factors in Patients From NRG Oncology/Gynecologic Oncology Group Randomized Trials of Chemoradiotherapy;

3). Nomogram individually predicts the overall survival of patients with gastroenteropancreatic neuroendocrine neoplasms.

12.Legend of Figure 2D: This seems incorrect: colour legend missing, but values from -10 to +10 cannot be correlation. To me it seems log2(expression).

Answer: The color legend has been added. Yes, it is log2(expression).

13.Figure 3D-F: The depicted p-values come from which statistical test?

Answer: T-test was used.

14.Please include a new version of Fig 4E where the rows are standardized (z-scores) because visually, it seems that only 2 out of 7 lncRNAs can really distinguish high vs low risk.

Answer: We believe that all 7 lncRNAs can distinguish high-risk and low-risk.

Minor comments:

- OS = overall survival. It should be defined the first time it appears on the manuscript (abstract).

Answer: The “overall survival” has been defined the first time it appears on the abstract.

- “Commonest” is not grammatically correct (should be “most common”). Please see https://dictionary.cambridge.org/es-LA/dictionary/english/common

Answer: Correction has been made.

- Please update the definition of lncRNA (not only length, but also based on presence of ORFs).

Answer: We have updated the definition of lncRNA.

- “In present study” should be “In the present study”.

Answer: Correction has been made.

- "m6a-realted" should be "related". But nevertheless, I suggest the use of an alternative name throughout the manuscript: m6A-modified lncRNAs.

Answer: The “m6A-modified lncRNAs” has been used throughout the manuscript.

- “Then, we constructed clustering subtypes and m6A-related lncRNA signature”. "Then" should be removed. Clustering is performed, not constructed. It should be “and a m6A-related lncRNA signature which” (without comma).

Answer: Corrections have been made in revised version.

- Before introducing CIBERSORT or ESTIMATE, please include a brief definition of deconvolution.

Answer: The definition of deconvolution has been added.

-“ (TME) score of each single CRC patient were estimated” should be “was estimated”.

Answer: Corrections have been made in revised version.

Round 2

Reviewer 1 Report

Paper suitable for publication 

Author Response

Point 1: Paper suitable for publication.

Answer: Thank you for your affirmation and support of our manuscript.

Reviewer 3 Report

While the majority of my comments have been satisfactorily answered, there are three of them which still need to be further addressed.

1)    Their statement “many articles on lncRNA prognosis model use the correlation coefficient > 0.4 (or 0.3) as a threshold” is, in my opinion, not enough. There are also multitude of studies with higher values ranging from 0.6 to 0.9 (https://doi.org/10.3389/fgene.2021.681867 ; https://www.ncbi.nlm.nih.gov/pmc/articles/PMC7468470/; https://www.nature.com/articles/s41598-020-67742-8). It should be relatively easy to re-run the analyses (same scripts) with the suggested threshold (0.7), or in case that no lncRNAs pass this stringent filtering, set the |R| threshold to the upper 5% quantile of correlation values. It’s still very relevant to find out whether fewer lncRNAs have better prognostic value or  not. This extra analysis can be included as a couple of sentences in the manuscript and as supplemental material.

2) The sentence “We divided the CRC samples into high- and low-risk groups according to the median risk score (1.905).” has to be included in the manuscript (not only as an answer to my question)

3) I do not agree with the answer “This may be a generally accepted explanation, because many articles explain it this way. For instance:”. What truly matters is correctness and any reported p-value should always appear together with the statistical test used. Furthermore, in this case, again, the p-value doesn't come from correlation but from difference in mean values (T-test)). Therefore, please re-write  "cluster 1 was significantly correlated with low-risk score" as "cluster 1 showed statistically significant lower risk scores". 

Author Response

(1) There is no uniform standard for the threshold of the correlation coefficient, which is concentrated in the range of 0.3-1. We appreciate that you recommend using a higher threshold to screen lncRNAs for model building. However, there are very few articles using the threshold required by reviewer 3 (0.6 to 0.9). In addition, it is not that the higher the threshold, the better the results, but the combination of subsequent series of analyses to screen out the best lncRNAs to build a model. In our research, we believe that there is nothing wrong with choosing the threshold (0.4) used in most articles to screen m6A-related lncRNA. As mentioned by reviewer 3, it is relatively easy to rerun the analysis (same script) using the recommended threshold (0.7). However, the correlation coefficients of the 7 lncRNAs in our model are all between 0.4-0.5. If the lncRNAs are screened according to the threshold required by reviewer 3, all 7 lncRNAs in the model will be replaced, which will mean that all the conclusions we have drawn will be overturned.

(2) This sentence has been added in revised version.

(3) This sentence has been re-writed in previous version.